

# Normal modes in thermal AdS via the Selberg zeta function

Victoria L. Martin[*] and Andrew Svesko[†]

Department of Physics, Arizona State University, Tempe, Arizona 85287, USA

[*] victoria.martin.2@asu.edu, [†] asvesko@asu.edu

## Abstract

The heat kernel and quasinormal mode methods of computing 1-loop partition functions of spin $s$ fields on hyperbolic quotient spacetimes $\mathbb{H}^3/\mathbb{Z}$ are related via the Selberg zeta function. We extend that analysis to thermal $\text{AdS}_{2n+1}$ backgrounds, with quotient structure $\mathbb{H}^{2n+1}/\mathbb{Z}$. Specifically, we demonstrate the zeros of the Selberg function encode the normal mode frequencies of spin fields upon removal of non-square-integrable modes. With this information we construct the 1-loop partition functions for symmetric transverse traceless tensors in terms of the Selberg zeta function and find exact agreement with the heat kernel method.



# 1 Introduction

In Euclidean quantum gravity the main object of interest is the partition function

$$Z = \int Dg D\phi \; e^{-S_E(g,\phi)/\hbar} \,, \tag{1}$$

where $g$ is the dynamical metric and $\phi$ represents all other matter fields. The leading order quantum effects are captured by the 1-loop partition function $Z_\phi^{(1)}$. For a free field $\phi$ on a gravitational background $\mathcal{M}$, computing $Z_\phi^{(1)}$ involves calculating functional determinants of kinetic operators $\nabla_{\phi,\mathcal{M}}^2$. This perturbative approach is useful when finding quantum corrections to black hole entropy [1–3] and holographic entanglement entropy [4].

There are several methods for computing functional determinants of kinetic operators. Two such methods we consider are known as the heat kernel method (c.f. [5]) and the quasinormal mode method [6]. We outline the heat kernel method in Appendix B and give a more extensive review of the quasinormal mode method in Section 2.2.

The quasinormal mode method [6,7] builds functional determinants of kinetic operators by exploiting the Weierstrass factorization theorem, which permits us to write a meromorphic function $Z^{(1)}(\Delta)$ as a product of its zeros and poles, up to an entire function $e^{\text{Poly}(\Delta)}$,

$$Z^{(1)}(\Delta) = e^{\text{Poly}(\Delta)} \frac{\prod_{\Delta_0} (\Delta - \Delta_0)^{d_0}}{\prod_{\Delta_p} (\Delta - \Delta_p)^{d_p}} \,. \tag{2}$$

Here $\Delta$ is a function of the mass $m$ of the field in question, and $d_0$ and $d_p$ are the degeneracies of the zeros $\Delta_0$ and poles $\Delta_p$, respectively. For examples we consider, $\Delta$ is the conformal dimension of the field theory operator dual to the bulk field in question. It was shown in [6,7] that $Z^{(1)}(\Delta)$ for a massive scalar field $\phi$ living on a thermal $\text{AdS}_{d+1}$ background may be expressed in terms of quasinormal modes[1] $\omega_*(\Delta)$:

$$Z^{(1)}(\Delta) = e^{\text{Poly}(\Delta)} \prod_{n,*} (\omega_n(T) - \omega_*(\Delta))^{-1} \,. \tag{3}$$

Here $*$ stands for a collection of additional quantum numbers, such as angular momentum, and $\omega_n(T)$ are the Matusubara frequencies of the thermal background at temperature $T$ arising from the condition that $\phi$ is regular in the Euclidean time coordinate. For stationary spacetimes, the Matsubara frequencies will generalize from $\omega_n(T) = 2\pi i n T$ to a function that depends on the angular momentum quantum number [8].

Recently the authors of [9] showed how to connect the heat kernel and quasinormal mode methods on the Bañados, Teitelboim and Zanelli (BTZ) black hole [10]. In particular, the two methods were formally related via the Selberg zeta function $Z_{\mathbb{Z}}(z)$ [11], a zeta function that is built entirely from the quotient structure of $\mathbb{H}^3/\mathbb{Z}$:

$$Z_{\mathbb{Z}}(z) = \prod_{k_1,k_2=0}^{\infty} \left[ 1 - e^{2ibk_1} e^{-2ibk_2} e^{-2a(k_1+k_2+z)} \right] \,. \tag{4}$$

In (4), $a$ and $b$ are geometric quantities[2]. Specifically, the authors of [9,12] showed that when the zeros of the Selberg zeta function $z^*$ are identified with $\Delta_s \pm \frac{isb}{a}$, and when the Selberg

---

[1]Quasinormal modes are eigenmodes of dissipative systems, such as those modes obeying infalling boundary conditions at a black hole horizon. The quasinormal mode method can be applied to spacetimes without horizons, such as thermal $\text{AdS}_N$. However, in these cases the quasinormal modes are replaced by normal modes as damping does not occur in such backgrounds.

[2]For the BTZ black hole $a = \pi r_+$ and $b = \pi|r_-|$, where $r_-$ and $r_+$ are the inner and outer horizon radii. For thermal $\text{AdS}_3$, $a = 1/(2T)$ and $b = \theta/2$, where $T$ is the temperature and $\theta$ is an angular potential.

integers $k_1$ and $k_2$ are appropriately recast via a relabeling inspired by scattering theory [13], the quasinormal mode frequencies $\omega_*$ are equal to the Matsubara frequencies $\omega_n(T)$:

$$\Delta_s \pm \frac{isb}{a} = z^* \Longleftrightarrow \omega_*(\Delta) = \omega_n(T) \,. \tag{5}$$

Condition (5) leads to the observation: if any two of (i) the Selberg zeta function, (ii) the Matsubara frequencies, or (iii) the quasinormal mode frequencies of a spacetime and field are known, then one can reconstruct the third. This observation provides a means of predicting quasinormal mode (or Matsubara) frequencies of fields on locally thermal AdS$_3$ spacetimes.

In this article we extend the results of [9,12] in two ways. First we confine ourselves to thermal AdS$_3$, and build the connection between heat kernel and quasinormal mode methods presented in [12] using instead the *normal* mode frequencies of general spin $s$ fields. We find that the relabelings of the Selberg integers $k_1$ and $k_2$ are the same as for the BTZ black hole [13], except with the thermal quantum number $n$ and angular quantum number $\ell$ switched. This connection is not surprising, in that the BTZ black hole and thermal AdS$_3$ are related via a modular transformation. However, this extension provides a testbed in which to study the ideas presented in [9,12]. Indeed, using the Selberg formalism we are able to "predict" the known normal modes of spin $s$ fields on thermal AdS$_3$.

Second, we extend [9,12] to higher dimensional thermal AdS$_{2n+1}$. To do this, we employ the higher dimensional generalization of (4), the Selberg zeta function on $\mathbb{H}^{2n+1}/\mathbb{Z}$ [11,14]:

$$Z_{\mathbb{Z}}(z) = \prod_{k_1,\dots,k_{2n}=0}^{\infty} \left(1 - e^{2ib_1 k_1} e^{-2ib_1 k_2} \dots e^{2ib_n k_{2n-1}} e^{-2ib_n k_{2n}} e^{-2a(k_1+\dots+k_{2n}+z)}\right) \,. \tag{6}$$

We conjecture an augmented relabeling for the integers $k_i$, generalizing [13]. We do this first for a complex scalar field. We then move to higher spin $s$ fields, and write an explicit formula for the 1-loop partition function for symmetric transverse traceless tensor fields on thermal AdS$_5$ in terms of the Selberg zeta function. We then discuss a generalization to AdS$_{2n+1}$.

Our note is organized as follows. We begin Section 2 with a brief review of the geometry and quotient structure of thermal AdS$_3$ (with non-zero angular potential), and demonstrate how the quasinormal mode method is used to calculate the 1-loop partition function for scalar fields on this spacetime. We then relate the zeros of the Selberg zeta function and normal mode frequencies of arbitrary spin fields on thermal AdS$_3$. In Section 3 we extend our analysis to thermal AdS$_{2n+1}$, both without and with non-zero angular potentials. Concluding remarks are given in Section 4. To keep the article self-contained, Appendix A gives an overview of the geometry of Euclidean AdS$_{2n+1}$, and Appendix B reviews basic elements of the group theoretic construction of the heat kernel on hyperbolic spaces [15,16].

## 2 Thermal AdS$_3$

### 2.1 Geometry of Thermal AdS$_3$

Anti-de Sitter space in three dimensions in global coordinates takes the form

$$ds^2 = L^2(-\cosh^2\rho \, dt^2 + d\rho^2 + \sinh^2\rho \, d\phi^2) \,, \tag{7}$$

where $L$ is the AdS radius. These coordinates have ranges $-\infty < t < \infty$, $0 < \rho < \infty$, and $0 < \phi < 2\pi$. Written in global coordinates, it is clear AdS$_3$ is static and axially symmetric, symmetries generated by the Killing vectors $H \equiv i\partial_t$ and $J \equiv i\partial_\phi$. These vectors can be used to define a notion of energy and angular momentum and define a pair of conserved charges.

We are interested in studying free quantum field theory on a fixed $AdS_3$ background. Upon quantization the vectors $H$ and $J$ become operators on the field theory Hilbert space such that the Hilbert space is organized into states of fixed energy and angular momentum. To properly define quantum field theory on an $AdS_3$ background we analytically continue $t \to -it_E$, to obtain the Euclidean $AdS_3$ metric

$$ds_E^2 = L^2(\cosh^2 \rho \, dt_E^2 + d\rho^2 + \sinh^2 \rho \, d\phi^2). \tag{8}$$

Euclidean $AdS_3$ is equivalent to the hyperbolic space $\mathbb{H}^3$. To obtain a thermal spacetime, we periodically identify the Euclidean time coordinate $t_E$ and the angular coordinate $\phi$ via

$$(t_E, \phi) \sim (t_E + \beta, \phi + \theta), \tag{9}$$

where $\beta$ is defined as the inverse temperature and $\theta$ is an angular potential. The identifications (9) allow us to recast the path integral (1) as a thermal partition function

$$Z(\beta, \theta) = \mathrm{tr} \, e^{-\beta H - i\theta J}. \tag{10}$$

Euclidean $AdS_3$ (8) with identifications (9) is known as thermal $AdS_3$. The identifications (9) generate the group $\mathbb{Z}$, and so thermal $AdS_3$ is topologically equivalent to the hyperbolic quotient $\mathbb{H}^3/\mathbb{Z}$. We can view $\mathbb{H}^3/\mathbb{Z}$ as a solid torus with a $T^2 \simeq S_\theta^1 \times S_\beta^1$ boundary and modular parameter $2\pi\tau = \theta + i\beta$ [5]. We can see from the $\rho = 0$ behavior of (8) that the Euclidean time circle $t_E \sim t_E + \beta$ is non-contractible, so $S_\theta^1$ fills in the solid torus.

## 2.2 Normal modes and the 1-loop partition function

Here we review the derivation of the 1-loop partition function of a spin-$s$ field living on thermal $AdS_3$ via the quasinormal mode method [6]. Our discussion is slightly more general than the one provided in [6] as we consider arbitrary spin-$s$ fields and $\theta \neq 0$. For concreteness we first consider a massive complex scalar field $\varphi$ of mass $m$.

The chief idea of the quasinormal mode method is to assume $Z^{(1)}$ is a meromorphic function of some mass parameter $\Delta$, and then analytically continue this mass parameter to the complex plane. For a scalar field the mass parameter is the conformal dimension

$$\Delta = 1 + \sqrt{1 + (mL)^2} \tag{11}$$

of the conformal field theory operator dual to $\varphi$.

If $Z^{(1)}(\Delta)$ is meromorphic in $\Delta$, we may use Weierstrass's factorization theorem (2) and express the 1-loop partition function as a product over its zeros and poles up to a entire function[3] $\mathrm{Poly}(\Delta)$. Since $Z_{\mathrm{scalar}}^{(1)} \propto (\det \nabla^2)^{-1}$, it has no zeros but will have a pole whenever $\nabla^2$ has a zero mode. Zero modes occur when $\Delta$ is tuned such that the Klein-Gordon equation

$$\left(-\nabla^2 + \frac{\Delta_{n,*}(\Delta_{n,*} - 2)}{L^2}\right)\varphi = 0 \tag{12}$$

has a smooth, single-valued solution $\varphi = \varphi_{*,n}$ in Euclidean signature which obeys the asymptotic boundary conditions, (14). Here $n$ labels the mode number in the Euclidean time direction and $*$ represents all other quantum numbers. The associated $\Delta$ for which $\varphi_{*,n}$ solve the Klein-Gordon equation are denoted $\Delta_{*,n}$. Thus, poles in $Z^{(1)}(\Delta)$ occur when $\Delta = \Delta_{*,n}$.

---

[3]As described in [6], $\mathrm{Poly}(\Delta)$ hides UV divergences. For example, a scalar field in AdS will contribute its zero point energy, $\sum_{\kappa=0}^{\infty}(\kappa + 1)\frac{\kappa + \Delta}{2T}$, to $\mathrm{Poly}(\Delta)$. To have a complete accounting of the 1-loop partition function we must determine $\mathrm{Poly}(\Delta)$. We are not interested in this divergent term and will often drop it from our calculations. $\mathrm{Poly}(\Delta)$ is fixed by imposing the correct large $\Delta$ behavior and using the heat kernel coefficients of the Laplacian $\nabla^2$ as in [17]. The term $\mathrm{Poly}(\Delta)$ is proportional to the volume of $\mathbb{H}^3/\mathbb{Z}$.

It was first realized in [6] that for scalar fields on AdS black hole backgrounds, the (anti)quasinormal modes are Lorentzian modes that are purely (out)ingoing at the horizon, and satisfy the asymptotic boundary conditions. Both conditions can only be satisfied at a set of discrete frequencies $\omega_*(\Delta)$, and setting $\Delta = \Delta_{*,n}$ is equivalent to setting the black hole quasinormal frequencies to the Matsubara frequencies $\omega_n(T)$ arising from the Euclidean periodicity condition, $\omega_*(\Delta_{*,n}) = \omega_n$. The 1-loop partition function is therefore computed using (3). Despite AdS not having a horizon, the quasinormal method of computing 1-loop determinants can nonetheless be applied in this context. Because there is no horizon, the key change is that there is no dissipation, and so the *quasi*normal frequencies $\omega_*$ become *normal* frequencies, and that there is no sensible difference between ingoing and outgoing modes.

The exact normal frequencies[4] for scalar fields are known [6, 19] and can be written as:

$$\omega_*(\Delta) = \pm \frac{(2p + |\ell| + \Delta)}{L} \, , \tag{13}$$

where $p = 0, 1, 2, \dots$ is the radial quantum number and $\ell$ is the angular momentum quantum number, $\ell \in \mathbb{Z}$. The mode frequencies (13) are found by solving the Klein-Gordon equation and imposing Dirichlet boundary conditions. From here on we will set $L = 1$.

The Matsubara frequencies are found via the large $\rho$ limit of the solution to the Klein-Gordon equation after imposing that $\varphi$ is periodic under identifications (9). The large $\rho$ limit of $\varphi$ is [19]

$$\varphi(\rho, t, \phi) \sim (\sinh \rho)^{-\Delta} e^{-i\omega t + i\ell\phi} \, . \tag{14}$$

Upon Wick rotation $t \to -i t_E$ and demanding that $\varphi$ be periodic under (9) sets the frequency $\omega$ to a particular form $\omega_n(T)$,

$$\omega_n(T) = 2\pi i n T - i\ell\theta T \, . \tag{15}$$

Here the thermal integer $n$ ranges over all integers. The frequencies $\omega_n(T)$ in (15)[5] are known as the Matsubara frequencies with a shift by a chemical potential $\mu = -\theta T$.

We can now construct the 1-loop determinant for a scalar field on thermal $\text{AdS}_3$ with $\theta \neq 0$. Substituting the normal mode frequencies (13) and Matsubara frequencies (15) into the expression for the 1-loop partition function over its poles (3), we find:

$$
\begin{aligned}
\frac{Z^{(1)}(\Delta)}{e^{\text{Poly}(\Delta)}} &= \prod_{n,*} (\omega_n(T) - \omega_*(\Delta))^{-1} \\
&= \prod_{n>0, p\geq 0, \ell} [(\omega_n(T) - \omega_{p,\ell}(\Delta))(\omega_{-n}(T) - \omega_{p,\ell}(\Delta))]^{-1} \\
&= \prod_{p\geq 0, \ell} \left(1 - e^{-\beta(2p + \Delta + |\ell|) - i\ell\theta}\right)^{-1} \, .
\end{aligned}
\tag{16}
$$

To get to the second line we cast the product over $n \in \mathbb{Z}$ into a single product over $n > 0$, absorbing any UV divergent pieces into the $\text{Poly}(\Delta)$ contribution, and moving to the third line regulated the product over $n$ using the identity $\prod_{n>0}\left(1 + \frac{x^2}{n^2}\right) = \frac{e^{\pi x}}{\pi x}(1 - e^{-2\pi x})$, again absorbing any UV divergent contribution into the entire function $\text{Poly}(\Delta)$.

---

[4]The $\text{AdS}_3$ normal frequencies can be obtained from the BTZ quasinormal frequencies via the identifications, $r_- \to 0$, and $r_+ \to \pm iL$ where $L$ is the AdS length scale, and $r_\pm$ are the outer and horizon radii. Correspondingly, $T_{\text{BTZ}} \to \mp 2\pi i L$. These are the same changes which transform the (Lorentzian) rotating BTZ metric in Boyer-Lindquist form with ADM mass and angular momentum $M = -1$ and $J = 0$, respectively, into (Lorentzian) $\text{AdS}_3$ in global coordinates [18].

[5]Using the modular transformation $\tau_{\text{AdS}_3} = -1/\tau_{\text{BTZ}}$ it is straightforward to show the Matsubara frequencies of thermal $\text{AdS}_3$ with $\theta \neq 0$ transform into the Matsubara frequencies for the Euclidean BTZ black hole with rotation (e.g., take Eqn. (2.8) of [8], rewrite the left and right temperatures $T_L$ and $T_R$ in terms of $\tau_{\text{BTZ}}$ and then perform the modular transformation).

We recast the product over $\ell \in \mathbb{Z}$ into one over $\ell \geq 0$, introducing a degeneracy factor $D_\ell^{(1)}$:

$$Z^{(1)}(\Delta) = e^{\text{Poly}(\Delta)} \prod_{p,\ell=0}^{\infty} \left[ (1 - (q\bar{q})^{p+\Delta/2} q^\ell)(1 - (q\bar{q})^{p+\Delta/2} \bar{q}^\ell) \right]^{-D_\ell^{(1)}} . \qquad (17)$$

Here $D_0^{(1)} = 1$ and $D_{\ell>0}^{(1)} = 2$, and $q = e^{2\pi i \tau} = e^{i(\theta + i\beta)}$ and $\bar{q} = e^{-2\pi i \bar{\tau}} = e^{-i(\theta - i\beta)}$. When we set $\theta = 0$ we recover the expression of the 1-loop partition function found in [6].

Taking the logarithm and evaluating the sums over $p$ and $\ell$, we arrive at

$$\log Z^{(1)}(\Delta) - \text{Poly}(\Delta) = 2 \sum_{k=1}^{\infty} \frac{|q|^{k\Delta}}{k|1 - q^k|^2} . \qquad (18)$$

Up to the entire function $\text{Poly}(\Delta)$, our expression (18) matches the 1-loop determinant $-\log \det(-\nabla^2 + m^2)$ of a scalar field on thermal $\text{AdS}_3$ found using the heat kernel method [15] (set $s = 0$ and $n = 1$ in (80)). The expression (18) is also the 1-loop determinant for the BTZ black hole because it is invariant under the modular transformation $\tau_{\text{AdS}_3} \to -1/\tau_{\text{BTZ}}$.

## 2.3 Normal Modes from Selberg Zeros and Higher Spin

In [9, 12] the authors formally connected the heat kernel and quasinormal mode methods using the Selberg zeta function. The Selberg zeta function $Z_\Gamma$ is a zeta function that is built entirely from the geometry of the hyperbolic quotient $\mathbb{H}^3/\Gamma$, where $\Gamma$ is a discrete subgroup of $SL(2, \mathbb{C})$. For example, $Z_\Gamma$ for the quotient space $\mathbb{H}^3/\mathbb{Z}$ is [13]

$$Z_{\mathbb{Z}}(z) = \prod_{k_1,k_2=0}^{\infty} \left[ 1 - e^{2ibk_1} e^{-2ibk_2} e^{-2a(k_1+k_2+z)} \right] . \qquad (19)$$

The parameters $a$ and $b$ are related to the identifications (9) of thermal $\text{AdS}_3$; specifically, $2a = \beta$ and $2b = \theta$. The zeros $z^*$ of the Selberg zeta function (19) are

$$z^* = -(k_1 + k_2) + \frac{i\theta}{\beta}(k_1 - k_2) + \frac{2\pi iN}{\beta} , \qquad (20)$$

where $N \in \mathbb{Z}$.

One-loop determinants can be recast in terms of the Selberg zeta function [9, 13, 20], e.g., for a scalar, $\log \det \nabla^2_{s=0} = 2 \log Z_\Gamma(\Delta)$. It is interesting to see what happens when we set the argument of the Selberg zeta function, $\Delta$, to the zeros (20):

$$\Delta + k_1 + k_2 = \frac{2\pi iN}{\beta} + \frac{i\theta}{\beta}(k_1 - k_2) . \qquad (21)$$

Notice that when we suggestively relabel the integers $k_1, k_2$ and $N$ such that

$$k_1 + k_2 = 2p + |\ell| , \quad k_1 - k_2 = \pm\ell , \quad N = \mp n , \qquad (22)$$

we find that (21) becomes

$$\mp(2p + |\ell| + \Delta) = (2\pi in - i\ell\theta)T . \qquad (23)$$

That is, tuning the conformal dimension $\Delta$ to the zeros $z^*$ of the Selberg zeta function gives us the condition that $\omega_n(T) = \omega_*(\Delta)$.

The relabeling of integers $k_1, k_2$ is not ad hoc. Rather, (22) comes from spectral theory on the hyperbolic quotient $\mathbb{H}^3/\mathbb{Z}$ [13], where $k_1, k_2$ are repackaged into new integers $p \geq 0$ and

$\ell \in \mathbb{Z}$ such that the zeros of the Selberg zeta function coincide with the poles of the so-called scattering operator $\Delta_\Gamma$, i.e., the positive Laplacian acting on the Hilbert space.

The observation that $\Delta = z^*$ leads to $\omega_n(T) = \omega_*$ was noted in [9], and generalized to include higher spin fields in [12] in the context of a rotating BTZ background. Our above analysis for thermal AdS$_3$ is largely the same as the one performed for the BTZ black hole [9], however, there are some key differences, specifically the physical interpretation of the integers appearing in the redefinition (22). For the BTZ black hole, the relabeling of $k_1, k_2$ is

$$k_1 + k_2 = 2p + |n|, \quad k_1 - k_2 = \pm n, \quad N = \mp \ell. \tag{24}$$

Comparing to the relabeling for thermal AdS$_3$ (22), we observe that the roles of the thermal integer $n$ and angular momentum quantum $\ell$ are swapped. This is reminiscent of the topology of each of these spacetimes: the thermal time circle for the BTZ black hole is contractible, while it is non-contractible for thermal AdS$_3$. Therefore, we see that the interpretation of $k_1 - k_2$ and $N$ are linked to spacetime topology, and signals the fact that the BTZ black hole and thermal AdS$_3$ are related via a modular transformation.

It was also emphasized in [9] that given knowledge of any two of (i) Matsubara frequencies, (ii) (quasi)normal mode frequencies, and (iii) the zeros of the Selberg zeta function, the third can be constructed. This provides us a means of predicting the normal modes of a field on thermal AdS$_3$. Let us use this predictive power to uncover the normal mode frequencies for an arbitrary spin-$s$ field in a thermal AdS$_3$ background.

We begin by considering spin-$s$ bosonic fields of mass $m_s$. The 1-loop determinant can be cast as a product of Selberg zeta functions [9, 12, 20]

$$\log \det(-\nabla^2_{(s)} + m_s^2)_{s \in \mathbb{Z}^+} = \log\left[ Z_\Gamma\left(\Delta_s + \frac{is\theta}{\beta}\right) \cdot Z_\Gamma\left(\Delta_s - \frac{is\theta}{\beta}\right)\right], \tag{25}$$

where $\Delta_s = 1 + \sqrt{s + 1 + m_s^2}$ is the conformal dimension of the dual CFT$_2$ operator [21]. At this point the arguments $\Delta_s \pm \frac{is\theta}{\beta}$ are perhaps unmotivated, but we will shed light on them shortly. Setting the arguments $\Delta_s + \sigma_\Delta \frac{is\theta}{\beta}$, where we use $\sigma_\Delta$ to denote the $\pm$ sign, to the zeros (20) leads to:

$$\Delta_s + k_1 + k_2 = \frac{i\theta}{\beta}(k_1 - k_2 - \sigma_\Delta s) + \frac{2\pi iN}{\beta}. \tag{26}$$

Then, motivated by the relabeling (22) of [13], and using the form of the Matsubara frequencies (15), we may write down

$$k_1 + k_2 = 2p + |\ell - \sigma_s s|, \quad k_1 - k_2 = -\sigma_\Delta \sigma_s \ell + \sigma_\Delta s, \quad N = \sigma_\Delta \sigma_s n, \tag{27}$$

where $\sigma_s = \pm 1$. For example, when $\sigma_\Delta = -1$, i.e., considering the argument $\Delta_s - \frac{is\theta}{\beta}$, and selecting $\sigma_s = -1$, we have

$$k_1 + k_2 = 2p + |\ell + s|, \quad k_1 - k_2 = -(\ell + s), \quad N = n, \tag{28}$$

such that (26) becomes

$$2p + |\ell + s| + \Delta_s = \omega_n(T). \tag{29}$$

This allows us to read off the normal mode frequency $\omega_*(\Delta_s) = 2p + |\ell + s| + \Delta_s$.

Collectively, the relabeling (27) substituted into (26) gives the condition $\omega_*(\Delta_s) = \omega_n(T)$ where we identify the normal mode frequencies of a spin-$s$ boson in thermal AdS$_3$

$$\omega_* = \pm(2p + |\ell \pm s| + \Delta_s). \tag{30}$$

Our relabeling (27) can be understood as a generalization of the redefinitions of integers $k_1$ and $k_2$ from [13].

A similar generalized relabeling was found for spin-$s$ bosons on the rotating BTZ background in [12], where the relabeling led to $\omega_n = \omega_*(\Delta_s)$ for only square-integrable Euclidean zero modes. More specifically, the Euclidean zero modes of higher spin fields will be non-square integrable for specific low lying values of $p$, a consequence of imposing regularity on the zero modes. In order to maintain regularity, one removes these non-square integrable modes, leading to the condition $\omega_n = \omega_*(\Delta_s)$. For example, in the case of a massive spin-2 field $h_{\mu\nu}$, Euclidean solutions $h_{\mu\nu}^{(\lambda)}$ are required to satisfy the following integrability condition [8]:

$$
\int d^3x \sqrt{g} \, g^{\mu\nu} g^{\rho\sigma} h_{\mu\rho}^{(\lambda)} h_{\nu\sigma}^{(\lambda')*} = \delta)\lambda - \lambda') \,, \tag{31}
$$

with $\lambda$ an eigenvalue. There will be non-integrable solutions at the radial coordinate singularity in the BTZ background in global coordinates. To avoid such non-integrable solutions, thereby maintaining regularity, the range over the thermal integer $n$ is restricted for Euclidean solutions with particular low-lying values of the radial quantum number $p$, where the remaining Euclidean solutions are referred to as "good" Euclidean zero modes (see, e.g., Appendix B of [8] for more details). Consequently, it was further shown in [8] that the conditions $\omega_n = \omega_*(\Delta_s)$ are equivalent to $\omega_n = \omega_*^{\text{scalar}}\left(\Delta_s + \sigma_\Delta \frac{is\theta_{\text{BTZ}}}{\beta_{\text{BTZ}}}\right)$, where $\sigma_\Delta$ depends on the sign of $m_s$, i.e., $\sigma_\Delta = -1$ corresponds to $m_s > 0$.

Since the Euclidean BTZ black hole is related to thermal AdS$_3$ via a modular transformation, higher spin fields in thermal AdS$_3$ will likewise have unphysical non-square integrable Euclidean zero modes for particular low lying values of $p$, upon adjusting the thermal integer $n$. The removal of these modes leads to the our conditions $\omega_n = \omega_*(\Delta_s)$ summarized by (26) with the relabeling (27). As such, we also have that arguments of the Selberg zeta functions in (25) come from removing the non-square integrable zeros modes and reexpressing $\omega_n = \omega_*(\Delta_s)$ for scalar fields with $\Delta \to \Delta_s + \sigma_\Delta \frac{is\theta}{\beta}$. Indeed, the 1-loop partition function for a spin-$s$ boson can be written as

$$
\begin{aligned}
Z_s^{(1)}(\Delta_s) &= -Z_\Gamma\left(\Delta_s + \frac{is\theta}{\beta}\right) Z_\Gamma\left(\Delta_s - \frac{is\theta}{\beta}\right) \\
&= Z^{(1)}\left(\Delta_s + \frac{is\theta}{\beta}\right) Z^{(1)}\left(\Delta_s - \frac{is\theta}{\beta}\right) \,,
\end{aligned} \tag{32}
$$

where $Z^{(1)}(\Delta)$ is the 1-loop partition function for a scalar field (17) without Poly$(\Delta)$.

Lastly, we note that our method works equally well for spin-$s$ fermions. The 1-loop determinant for spin-$s$ fermions with anti-periodic boundary conditions is

$$
\log \det(-\nabla_{(s)}^2 + m_s^2)_{s \in \mathbb{Z}_{\frac{1}{2}}^+} = \log\left[Z_\Gamma\left(\Delta_s + \frac{is\theta}{\beta} + \frac{i\pi}{\beta}\right) \cdot Z_\Gamma\left(\Delta_s - \frac{is\theta}{\beta} + \frac{i\pi}{\beta}\right)\right] \,. \tag{33}
$$

The anti-periodic boundary conditions along the Euclidean circle force $n \to n + 1/2$. As in the case of spin-$s$ fermions on a BTZ background [12], we again find that anti-periodic boundary conditions along the $\phi$ cycle are imposed upon us, such that $\ell \to \ell + 1/2$. Our spin-$s$ fermion result is the same as the spin-$s$ boson one (27), except with $n \to n + 1/2$ and $\ell \to \ell + 1/2$.

In summary, the zeros of the Selberg zeta function encode the normal mode frequencies of arbitrary spin-$s$ fields which propagate on thermal AdS$_3$. Moreover, setting the arguments $\Delta_s \pm \frac{is\theta}{\beta}$ (for bosons) or $\Delta_s \pm \frac{is\theta}{\beta} + \frac{i\pi}{\beta}$ (for fermions) equal to the zeros of the Selberg zeta function leads to the condition the Matsubara frequencies are aligned with the normal mode frequencies:

$$
\Delta_s \pm \frac{is\theta}{\beta} = z^* \Longleftrightarrow \omega_n = \omega_*(\Delta_s) \,. \tag{34}
$$

# 3 Extending to Thermal AdS$_{2n+1}$

We now extend the analysis presented in Section 2 to thermal AdS$_{2n+1}$. The geometry of AdS$_{2n+1}$ is reviewed in Appendix A. Generally, thermal AdS$_{2n+1}$ can be viewed as the quotient space $\mathbb{H}^{2n+1}/\mathbb{Z}$. The quotient structure arises from the periodic identification of the Euclidean time coordinate $t_E$ and the remaining angular coordinates $\{\phi_1, ..., \phi_{n+1}\}$ being identified via

$$(t_E, \phi_1, ..., \phi_n) \sim (t_E + \beta, \phi_1 + \theta_1, ..., \phi_n + \theta_n), \tag{35}$$

where $\beta$ is the inverse temperature and $\theta_i$ are angular potentials.

Written in global coordinates (68), we see AdS$_{2n+1}$ has symmetries generated by the Killing vectors $H \equiv i\partial_t$ and $J_i \equiv i\partial_{\phi_i}$, defining phase space charges associated with energy $H$ and angular momenta $J_i$. Upon quantization, the vectors $H$ and $J_i$ organize the field theory Hilbert space into states of fixed energy and angular momenta. Field theory quantities are computed using the canonical ensemble partition function $Z(\beta, \theta_i)$,

$$Z(\beta, \theta_i) = \mathrm{tr}\, e^{-\beta H - i \sum_{j=1}^n \theta_j J_j}, \tag{36}$$

generalizing the thermal partition function in AdS$_3$, (10).

Evaluating the partition function (36) is equivalent to calculating a Euclidean path integral on $\mathbb{H}^{2n+1}/\mathbb{Z}$. The leading order quantum effects are quantified by the 1-loop partition function $Z^{(1)}$. The 1-loop partition function for complex scalar fields on thermal AdS$_{d+1}$ without angular potentials was computed using the quasinormal mode method in [6]. The 1-loop determinant for STT tensor fields on thermal AdS$_{2n+1}$ with non-zero angular potentials was constructed using the heat kernel method [16, 22] (also reviewed in Appendix B).

Here we explore the connection between the 1-loop partition function, the Selberg zeta function on $\mathbb{H}^{2n+1}/\mathbb{Z}$, and the normal modes of STT tensor fields on thermal AdS$_{2n+1}$, both with and without angular potentials. We begin by considering STT tensor fields on thermal AdS$_{2n+1}$ with $\theta_i = 0$. The remaining discussion will mirror the presentation in Section 2, where we begin with a scalar field on thermal AdS$_{2n+1}$ when $\theta_i \neq 0$. We will then consider higher spin field partition functions, where we explicitly write the 1-loop partition function in AdS$_5$ in terms of a higher dimensional Selberg zeta function.

## 3.1 Thermal AdS$_{d+1}$ ($\theta_i = 0$)

It is straightforward to generalize the complex scalar field 1-loop partition function in [6] to include symmetric, transverse, traceless (STT) tensor fields of spin-$s$[6]

$$Z_{(s)}^{(1)}(\Delta_s) = e^{\mathrm{Poly}(\Delta)} \prod_{p,\ell \geq 0} \left[ \left(1 - e^{-\beta(2p+\ell+\Delta_s)}\right)^{d_s(2-\delta_{s,0})} \right]^{-2D_\ell^{(d-1)}}, \tag{37}$$

where $\Delta_s$ is the conformal dimension $\Delta_s = \frac{d}{2} + \sqrt{s + \frac{d^2}{4} + m_s^2}$. Here $d_s$ is the dimension (75) when $d+1$ is odd, and $D_\ell^{(d-1)}$ is the degeneracy of the $\ell$th angular momentum eigenvalue

$$D_\ell^{(d-1)} = \frac{2\ell + d - 2}{d - 2} \frac{(\ell + d - 3)!}{\ell!(d-3)!}. \tag{38}$$

Some degenerate cases of note include when $d = 1$ for which $D_0^{(0)} = D_1^{(0)} = 1$ and $D_{\ell>1}^{(0)} = 0$, while for $d = 2$, $D_0^{(1)} = 1$ and $D_{\ell>0}^{(1)} = 2$.

---

[6]By spin-$s$ we mean unitary irreducible representations of $SO(2n+1)$ under which the fields transform. We further restrict ourselves to symmetric transverse traceless representations of $SO(2n+1)$, the highest weight representations, which greatly simplifies our study. Such fields include bosons of spin-$s$.

Taking the logarithm of (37) yields

$$\log Z_s^{(1)}(\Delta_s) = d_s(2 - \delta_{s,0}) \sum_{k=1}^{\infty} \frac{2e^{-k\Delta_s\beta}}{k} \sum_{p=0}^{\infty} e^{-2pk\beta} \sum_{\ell=0}^{\infty} D_\ell^{(d-1)} e^{-k\ell\beta} \,, \tag{39}$$

where we ignore the Poly($\Delta_s$) term. Performing the sums of $p$ and $\ell$ we find, (check steps here)

$$\log Z_s^{(1)}(\Delta_s) = d_s(2 - \delta_{s,0}) \sum_{k=1}^{\infty} \frac{2}{k} \frac{e^{-\beta k(\Delta_s - d)}}{(1 - e^{k\beta})^d} \,, \tag{40}$$

matching the heat kernel result (78), with $\theta_i = 0$.

Interestingly, one can build the 1-loop partition function (40) in odd dimensions using multiple copies of the AdS$_3$ result, (18) (when $\theta = 0$). For example, consider thermal AdS$_5$. Replacing $p \to p_1 + p_2$ and $\ell \to \ell_1 + \ell_2$ where $\{p_i, \ell_i\} \in \mathbb{N}$, and $D_\ell^{(3)} \to D_{\ell_1}^{(1)} D_{\ell_2}^{(1)}$, we have

$$\begin{aligned}
\log Z_{s=0}^{(1)}(\Delta) &= \sum_{k=1}^{\infty} \frac{2e^{-k\Delta\beta}}{k} \left( \sum_{p_i=0}^{\infty} e^{-2p_i k\beta} \right)^2 \left( \sum_{\ell_1=0}^{\infty} D_{\ell_1}^{(1)} e^{-k\ell_1\beta} \right) \left( \sum_{\ell_2=0}^{\infty} D_{\ell_2}^{(1)} e^{-k\ell_2\beta} \right) \\
&= \sum_{k=1}^{\infty} \frac{2e^{-k\Delta\beta}}{k} \frac{1}{(1 - e^{-\beta k})^4} \,,
\end{aligned} \tag{41}$$

which matches the scalar result (40). We can likewise build the full 1-loop determinant in AdS$_{2n+1}$, where for each additional odd dimension, introduce another $p_i$ and $\ell_i$. For example, in the AdS$_7$ case let $p \to p_1 + p_2 + p_3$ and $\ell \to \ell_1 + \ell_2 + \ell_3$, and so forth[7]. We will show that introducing additional integers $p_i$ and $\ell_i$ is motivated from the higher dimensional Selberg zeta function.

### 3.2 Thermal AdS$_{2n+1}$ ($\theta_i \neq 0$)

**Scalar fields**

To compute the 1-loop partition function, we must have knowledge of the Matsubara frequencies $\omega_n$ and the normal mode frequencies for a scalar field on AdS$_{2n+1}$. The periodic identification (35) imposed on a scalar field $\varphi$ leads to a generalization of the Matsubara frequencies in thermal AdS$_3$ (15),

$$\omega_{\tilde{n}}(T) = 2\pi i \tilde{n} T - i \sum_{i=1}^{n} \ell_i \theta_i T \,, \tag{42}$$

where $\tilde{n} \in \mathbb{Z}$ is the thermal integer and $\ell_i \in \mathbb{Z}$ is the angular momentum quantum number with respect to each $\phi_i$ in the geometry (69).

The normal mode frequencies for scalar fields on AdS$_{2n+1}$ were calculated explicitly in [19] in terms of a single radial quantum number $p$ and angular momentum quantum number $\ell$. Let us instead use the Selberg zeta function to "predict" these normal mode frequencies, extending our algorithm developed for thermal AdS$_3$ to higher-dimensional thermal AdS.

The Selberg zeta function of $\mathbb{H}^{2n+1}/\mathbb{Z}$ is given by [11, 14]

$$Z_{\mathbb{Z}}(z) = \prod_{k_1,\ldots,k_{2n}=0}^{\infty} \left( 1 - e^{2ib_1 k_1} e^{-2ib_1 k_2} \ldots e^{2ib_n k_{2n-1}} e^{-2ib_n k_{2n}} e^{-2a(k_1 + \ldots + k_{2n} + z)} \right) \,, \tag{43}$$

---

[7]We can in fact build the 1-loop determinant in even dimensions in a similar way, but with one less copy of $p$. For example, in AdS$_4$, $p \to p$ and $\ell \to \ell_1 + \ell_2$; for AdS$_6$, $p \to p_1 + p_2$, and $\ell \to \ell_1 + \ell_2 + \ell_3$, etc.

where $2a$ is the length of the primitive closed geodesic and $e^{2ib_i}$ are the eigenvalues of a rotation matrix $A$ describing the rotation of nearby closed geodesics under the Poincaré recurrence map. In the context of thermal $AdS_{2n+1}$, we identify $2a = \beta$ and $2b_i = \theta_i$. The zeros of $Z_{\mathbb{Z}}(z)$ occur at the special value of $z^*$,

$$z^* = -\sum_{i=1}^{n}(k_{2i-1} + k_{2i}) + \frac{i}{a}\sum_{i=1}^{n} b_i(k_{2i-1} - k_{2i}) + \frac{iN\pi}{a} , \tag{44}$$

where $N \in \mathbb{Z}$.

Taking the logarithm and evaluating the resulting sums over integers $k_1, ..., k_{2n}$ leads to

$$\begin{aligned}
\log Z_{\mathbb{Z}}(z) &= -\sum_{k=1}^{\infty} \frac{e^{-2akz}}{k} \frac{1}{e^{-2ank}\prod_{i=1}^{n}|e^{2ak} - e^{2ikb_i}|^2} \\
&= -\sum_{k=1}^{\infty} \frac{e^{-2akz}}{k} \frac{1}{\prod_{i=1}^{n}|1-q_i^k|^2} ,
\end{aligned} \tag{45}$$

where we have introduced a 'modular' parameter for each angular potential,

$$2\pi\tau \equiv \theta_1 + i\beta , \quad 2\pi\tau' \equiv \theta_2 + i\beta , \quad 2\pi\tau'' \equiv \theta_3 + i\beta , ... \tag{46}$$

such that

$$q_1 \equiv e^{2\pi i\tau} = e^{i(\theta_1 + i\beta)} , \quad q_2 \equiv e^{2\pi i\tau'} = e^{i(\theta_2 + i\beta)} , ... \tag{47}$$

Notice the denominator matches the denominator of the Harish-Chandra character (77).

Setting $\Delta = z^*$ gives us

$$\Delta + \sum_{i=1}^{n}(k_{2i-1} + k_{2i}) = \frac{i}{\beta}\sum_{i=1}^{n}\theta_i(k_{2i-1} - k_{2i}) + \frac{2\pi iN}{\beta} . \tag{48}$$

Motivated by the relabeling (22) of $k_1$ and $k_2$ from thermal $AdS_3$, we consider the following relabeling

$$k_{2i-1} + k_{2i} = 2p_i + |\ell_i| , \quad k_{2i-1} - k_{2i} = \sigma_\ell \ell_i , \quad N = -\sigma_\ell \tilde{n} , \tag{49}$$

where $p_i$ is a non-negative integer, $\ell_i \in \mathbb{Z}$, and $\sigma_\ell$ is the sign of $\ell_i$. For example, in thermal $AdS_5$, we have that (49)

$$\begin{aligned}
k_1 + k_2 &= 2p_1 + |\ell_1| , \quad k_1 - k_2 = \sigma_\ell \ell_1 \\
k_3 + k_4 &= 2p_2 + |\ell_2| , \quad k_3 - k_4 = \sigma_\ell \ell_2 , \quad N = -\sigma_\ell \tilde{n} .
\end{aligned} \tag{50}$$

The sign $\sigma_\ell$ is written such that $\sigma_\ell = +1$ when $k_1 - k_2 = +\ell_1$ and $k_3 - k_4 = +\ell_2$, and $\sigma_\ell = -1$ when $k_1 - k_2 = -\ell_1$ and $k_3 - k_4 = -\ell_2$. We do not consider any mixture of signs, e.g., $k_1 - k_2 = +\ell_1$ and $k_3 - k_4 = -\ell_2$. This can also be accomplished by setting the sign of $\ell_1$, and then relabeling each subsequent $k_{2i-1} - k_{2i}$ to have the same sign as $\ell_1$.

Using our conjectured[8] relabeling (49) we arrive at

$$\pm(2p + |\ell_1| + ... |\ell_n| + \Delta) = 2\pi i\tilde{n}T - i\sum_{i=1}^{n}\ell_i\theta_i T , \tag{51}$$

---

[8]Unlike in the case of $AdS_3$, we are unaware of whether our relabeling (49) arises from identifying the zeros of the Selberg zeta function with the poles of the scattering operator $\Delta_\Gamma$ on $\mathbb{H}^{2n+1}/\mathbb{Z}$. Our higher dimensional relabeling is therefore a conjecture. It would be interesting to generalize the relationship observed in [13] to determine if (49) comes from studying scattering on $\mathbb{H}^{2n+1}/\mathbb{Z}$.

where $p \equiv p_1 + p_2 + ... + p_n$. Recognizing the right hand side as the Matsubara frequencies (42), we 'predict' the normal mode frequencies for a scalar field in $AdS_{2n+1}$ to be

$$\omega_* = \pm(2p + |\ell_1| + ... + |\ell_n| + \Delta) , \tag{52}$$

where we again find that tuning the conformal dimension $\Delta$ to the zeros of the Selberg zeta function, we uncover the condition $\omega_*(\Delta) = \omega_n(T)$.

As a consistency check, we can substitute the Matsubara (42) and normal mode frequencies (52) into our expression for the 1-loop partition function (3) and show that we reproduce the 1-loop partition function calculated using the heat kernel method (81). For concreteness, consider $AdS_5$. Following similar steps as in the $AdS_3$ case, (17), we obtain

$$\log Z_{s=0}^{(1)} = \frac{1}{2} \log \prod_{\substack{p_1,p_2, \\ \ell_1,\ell_2=0}}^{\infty} \left\{ (1 - (q_1\bar{q}_1)^{p_1+\frac{\Delta}{4}}(q_2\bar{q}_2)^{p_2+\frac{\Delta}{4}}q_1^{\ell_1}q_2^{\ell_2})(1 - (q_1\bar{q}_1)^{p_1+\frac{\Delta}{4}}(q_2\bar{q}_2)^{p_2+\frac{\Delta}{2}}q_1^{\ell_1}\bar{q}_2^{\ell_2}) \right.$$
$$\left. (1 - (q_1\bar{q}_1)^{p_1+\frac{\Delta}{4}}(q_2\bar{q}_2)^{p_2+\frac{\Delta}{4}}\bar{q}_1^{\ell_1}q_2^{\ell_2})(1 - (q_1\bar{q}_1)^{p_1+\frac{\Delta}{4}}(q_2\bar{q}_2)^{p_2+\frac{\Delta}{4}}\bar{q}_1^{\ell_1}\bar{q}_2^{\ell_2}) \right\}^{-D_{\ell_1}^{(1)}D_{\ell_2}^{(1)}} . \tag{53}$$

Or, using $(q_1\bar{q}_1) = (q_2\bar{q}_2) = e^{-2\beta}$ we may write

$$\log Z_{s=0}^{(1)} = \frac{1}{2} \log \prod_{\substack{p_1,p_2, \\ \ell_1,\ell_2=0}}^{\infty} \left\{ (1 - e^{-2\beta(p_1+p_2+\frac{\Delta}{2})}q_1^{\ell_1}q_2^{\ell_2})(1 - e^{-2\beta(p_1+p_2+\frac{\Delta}{2})}q_1^{\ell_1}\bar{q}_2^{\ell_2}) \right.$$
$$\left. (1 - e^{-2\beta(p_1+p_2+\frac{\Delta}{2})}\bar{q}_1^{\ell_1}q_2^{\ell_2})(1 - e^{-2\beta(p_1+p_2+\frac{\Delta}{2})}\bar{q}_1^{\ell_1}\bar{q}_2^{\ell_2}) \right\}^{-D_{\ell_1}^{(1)}D_{\ell_2}^{(1)}} . \tag{54}$$

Note that when we turn off the angular potentials, define $p_1 + p_2 \equiv p$, $|\ell_1| + |\ell_2| \equiv \ell$ and make the replacement $D_{\ell_1}^{(1)}D_{\ell_2}^{(1)} \to D_\ell^{(3)}$ we recover the logarithm of (37) for scalars. Taking the logarithm and evaluating the sums over integers $p_1, p_2, \ell_1,$ and $\ell_2$, we arrive to the 1-loop partition function computed using the heat kernel method (81) at $s = 0$.

The above procedure holds for the scalar field $AdS_{2n+1}$. Each higher dimension includes an additional $p_i$ and $\ell_i$, from which it is straightforward to show

$$\log Z_{s=0}^{(1)} = \frac{1}{2^{n-1}} \log \prod_{\substack{p_1,p_2,...,p_n \\ \ell_1,\ell_2,...,\ell_n=0}}^{\infty} \left\{ (1 - e^{-2\beta(p_1+p_2+...+p_n+\frac{\Delta}{2})}q_1^{\ell_1}q_2^{\ell_2}...q_n^{\ell_2}) \right.$$
$$\left. (1 - e^{-2\beta(p_1+p_2+...+p_n+\frac{\Delta}{2})}q_1^{\ell_1}q_2^{\ell_2}...\bar{q}_n^{\ell_n})...(1 - e^{-2\beta(p_1+p_2+...+p_n+\frac{\Delta}{2})}\bar{q}_1^{\ell_1}\bar{q}_2^{\ell_2}...\bar{q}_n^{\ell_n}) \right\}^{-D_{\ell_1}^{(1)}D_{\ell_2}^{(1)}...D_{\ell_n}^{(1)}} , \tag{55}$$

where ... implies the permutations $q_1 q_2 ... q_n$ and their complex conjugates corresponding to rewriting the products over $\ell_i$ to range from all integers to all non-negative integers. Evaluting the sum over $\ell_i$ and $p_i$ yields the $s = 0$ 1-loop partition function computed using the heat kernel method (80).

## Higher Spin

We now turn to building the 1-loop partition functions of STT spin-$s$ fields on thermal $AdS_{2n+1}$. This was accomplished using the heat kernel method in [16]. To use the quasinormal mode method, we need the Matsubara frequencies (42) and the normal mode frequencies for STT

tensor fields. We will follow our approach in Section 2 and uncover the normal mode frequencies using the zeros of the Selberg zeta function.

In the AdS$_3$ case, recall we first recast the 1-loop determinant for arbitrary spin-$s$ fields – found via the heat kernel method – in terms of a product of Selberg zeta functions. We then tuned the arguments of the Selberg zeta functions to their zeros and extended the relabeling of integers $k_1$ and $k_2$ from [13]. Our first task then is to rewrite the 1-loop partition function (80) in terms of Selberg zeta functions. For concreteness, we will consider first the AdS$_5$ case and then comment on general AdS$_{2n+1}$.

The Selberg zeta function for $\mathbb{H}^5/\mathbb{Z}$ is (45)

$$\log Z_{\mathbb{Z}}(z) = -\sum_{k=1}^{\infty} \frac{(q_1\bar{q}_1)^{kz/4}(q_2\bar{q}_2)^{kz/4}}{k} \frac{1}{|1-q_1^k|^2|1-q_2^k|^2} \, . \tag{56}$$

We can incorporate the character formula $\chi_s^{SO(4)}$ (82) appearing in (81) using the geometric series representation of $SU(2)$ characters, e.g.,

$$\frac{\sin[\alpha_+(s+1)]}{\sin(\alpha_+)} = \sum_{m=-s/2}^{s/2} e^{-2im\alpha_+} = e^{is\alpha_+} \sum_{m=0}^{s} e^{-2im\alpha_+} \, . \tag{57}$$

Using $\alpha_{\pm} = (\theta_1 \pm \theta_2)/2$, we have

$$\frac{\sin[k(s+1)\alpha_+]}{\sin(k\alpha_+)} \frac{\sin[k(s+1)\alpha_-]}{\sin(k\alpha_-)} = e^{isk\theta_1} \sum_{m,m'=0}^{s} e^{-ik\theta_1(m+m')} e^{-ik\theta_2(m-m')} \, , \tag{58}$$

leading to the 1-loop partition function for STT tensors in thermal AdS$_5$ in terms of Selberg zeta functions

$$\log Z_s^{(1)} = -(2-\delta_{s,0}) \sum_{m,m'=0}^{s} \log Z_{\Gamma}\left[\Delta_s + \frac{i\theta_1}{\beta}(m+m'-s) + \frac{i\theta_2}{\beta}(m-m')\right] . \tag{59}$$

For example, for spin-2 fields we have

$$\begin{aligned}
\log Z_2^{(1)} = -2\Bigg\{ &\log Z_{\Gamma}\left(\Delta_2 + \frac{2i\theta_1}{\beta}\right) + \log Z_{\Gamma}\left(\Delta_2 - \frac{2i\theta_1}{\beta}\right) + \log Z_{\Gamma}\left(\Delta_2 + \frac{2i\theta_2}{\beta}\right) \\
&+ \log Z_{\Gamma}\left(\Delta_2 - \frac{2i\theta_2}{\beta}\right) + \log Z_{\Gamma}\left(\Delta_2 + \frac{i\theta_1}{\beta} - \frac{i\theta_2}{\beta}\right) + \log Z_{\Gamma}\left(\Delta_2 - \frac{i\theta_1}{\beta} + \frac{i\theta_2}{\beta}\right) \\
&+ \log Z_{\Gamma}\left(\Delta_2 + \frac{i\theta_1}{\beta} + \frac{i\theta_2}{\beta}\right) + \log Z_{\Gamma}\left(\Delta_2 - \frac{i\theta_1}{\beta} - \frac{i\theta_2}{\beta}\right) + \log Z_{\Gamma}(\Delta_2)\Bigg\} .
\end{aligned} \tag{60}$$

We see that, just as in the AdS$_3$ case (32), the spin-$s$ 1-loop partition function on AdS$_5$ breaks into a product of scalar 1-loop partition functions,

$$Z_s^{(1)} = \prod_{m,m'=0}^{s} Z^{(1)}\left(\Delta = \Delta_s + \frac{i\theta_1}{\beta}(m+m'-s) + \frac{i\theta_2}{\beta}(m-m')\right) , \tag{61}$$

where $Z^{(1)}(\Delta)$ is given by (54). We expect the arguments of the Selberg zeta function (59) arise from an analysis similar to the AdS$_3$ set-up where we must remove the non-square integrable zero modes for particularly low lying values of $p_i$, resulting from a readjustment of the integers $\tilde{n}$. This expectation is based on our observation that the higher dimensional 1-loop partition functions can be constructed from copies of the three-dimensional result, where the removal of the non-integrable zero mode analysis has been completed [8]. The actual removal of the

non-integrable Euclidean zero modes in the case of higher spin fields in higher dimensions has not been done explicitly (as this would require a full analysis the quasinormal mode method for massive higher spin fields on thermal AdS$_{2n+1}$, which has not yet been accomplished). Using our prediction for the normal mode frequencies of higher spin fields (see (64)), it would be worthwhile to explicitly complete this analysis.

Setting the arguments of the Selberg zeta functions (59) to the zeros (44) for $n = 2$, we obtain

$$\Delta_s + (k_1 + k_2) + (k_3 + k_4) = \frac{i\theta_1}{\beta}\left[k_1 - k_2 - (m + m' - s)\right] + \frac{i\theta_2}{\beta}\left[k_3 - k_4 - (m - m')\right] + \frac{2\pi i N}{\beta},$$
(62)

with $m, m' = 0, 1, ..., s$. We now extend the relabeling presented in (50):

$$
\begin{aligned}
k_1 + k_2 &= 2p_1 + |\ell_1 + \sigma_\ell(m + m' - s)|, \quad k_1 - k_2 = \sigma_\ell \ell_1 + (m + m' - s), \\
k_3 + k_4 &= 2p_2 + |\ell_2 + \sigma_\ell(m - m')|, \quad k_3 - k_4 = \sigma_\ell \ell_2 + (m - m'), \quad N = -\sigma_\ell \tilde{n}.
\end{aligned}
$$
(63)

Since there are $(s + 1)^2$ combinations of $m$ and $m'$, we have $(s + 1)^2$ different relabelings of the pairs $k_1, k_2$ and $k_3, k_4$.

Collectively, the relabeling (63) gives

$$\pm(2p + |\ell_1 \pm (m + m' - s)| + |\ell_2 \pm (m - m')| + \Delta_s) = \omega_n(T),$$
(64)

where $p \equiv p_1 + p_2$. We are inclined to interpret the left hand side as the set of normal mode frequencies $\omega_*(\Delta_s)$, such that (64) becomes $\omega_*(\Delta_s) = \omega_n(T)$. Unsurprisingly, it is straightforward to confirm that substituting the $(s + 1)^2$ collection of normal mode frequencies into the 1-loop partition function (3) leads to the spin-$s$ result found via the heat kernel method. We expect that the normal mode frequencies displayed in (64) will arise from a spin-$s$ quasinormal mode analysis on thermal AdS$_{2n+1}$, whereupon we remove non-square integrable Euclidean zero modes. We leave this confirmation for future work.

Our study of 1-loop partition functions of higher spin on AdS$_5$ informs us of how to extend the result to AdS$_{2n+1}$. Specifically, the spin-$s$ 1-loop partition function $Z_s^{(1)}(\Delta_s)$ on AdS$_{2n+1}$ is a product of $d_s^{(2n-1)}$ scalar field 1-loop partition functions[9] $Z^{(1)}(\Delta)$ on AdS$_{2n+1}$, corresponding to the number of replacements made to $\Delta$. For example, in AdS$_3$ we found that for $s \neq 0$, the 1-loop partition function $Z_s^{(1)}(\Delta_s)$ is a product of two scalar field partition functions, such that $\Delta$ is replaced by $\Delta_s \pm is\theta_1/\beta$, and the magnitude of the integer coefficient to $i\theta_1/\beta$ matches the spin of the field (32), corresponding to $d_s^{(1)} = 2$ for any $s > 0$. Moreover, for AdS$_5$, the 1-loop partition function is the product of $(s + 1)^2$ scalar field partition functions, such that the magnitude of the integer coefficients in front of each $i\theta_i/\beta$ sum to the total spin of the field. For example, for $s = 2$, we replace $\Delta$ with each argument appearing in the Selberg zeta functions in (60).

Likewise, we may build $Z_s^{(1)}(\Delta_s)$ on AdS$_{2n+1}$ from a product of scalar field partition functions on AdS$_{2n+1}$, where the number of terms in the product correspond to the number of replacements to $\Delta$, where the magnitude of the integer coefficients in front of each $i\theta_i/\beta$ must sum to the spin of the field $s$. For example, for a spin-1 field on AdS$_7$, $Z_1^{(1)}(\Delta_1)$ is given by a product of six scalar field 1-loop partition functions corresponding to the replacements of $\Delta$ by $\Delta \to \Delta_1 \pm \frac{i\theta_i}{\beta}$, for $i = 1, 2, 3$ (using $d_s^{(5)} = (s + 3)(s + 2)^2(s + 1)/12$). For each higher dimension, the number of relabelings (63) will also go as $d_s^{(2n-1)}$.

---

[9]Or, equivalently, a product over Selberg zeta functions $Z_\Gamma(\Delta)$, where the number of terms in the product is given by $d_s^{(2n-1)}$ corresponding to the replacements of $\Delta$.

# 4  Discussion

We have extended the relationship between the heat kernel and quasinormal mode methods for computing 1-loop determinants, as explored in [9,12], to $\mathbb{H}^{2n+1}/\mathbb{Z}$, i.e., thermal $\text{AdS}_{2n+1}$. First we considered arbitrary spin-$s$ fields propagating on a thermal $\text{AdS}_3$ background and showed that by tuning the zeros of the Selberg zeta function to conformal dimensions $\Delta_s$, we arrive at the condition that the normal modes must be identified with the Matsubara frequencies of thermal $\text{AdS}_3$. Comparing to our previous work with the BTZ black hole, we observed the relabeling of the integers $k_1, k_2$ depends on which circle of the solid torus $\mathbb{H}^3/\mathbb{Z}$ is filled in. We then extended our analysis on $\mathbb{H}^3/\mathbb{Z}$ to higher dimensional thermal $\text{AdS}_{2n+1}$ for arbitrary STT tensor fields. These generalizations to [9,12] allowed us to derive the normal modes for STT tensors on thermal $\text{AdS}_{2n+1}$ with angular potentials, using the zeros of the Selberg zeta function. With the normal modes we verified for consistency that the sum over radial and angular momentum quantum numbers $p$ and $\ell_i$ build up the global characters of $SO(2n+1, 1)$, and the sum over images arises from taking the logarithm of the functional determinant, matching the heat kernel results found in [16]. Finally, we developed an algorithm for recasting the higher spin field 1-loop partition functions as a product of Selberg zeta functions on $\mathbb{H}^{2n+1}/\mathbb{Z}$.

There are multiple generalizations and potential applications of our work. For example, we mostly focused on odd-dimensional hyperbolic quotients $\mathbb{H}^{2n+1}/\mathbb{Z}$. In the case of higher spin fields, even dimensional quotients $\mathbb{H}^{2n}/\mathbb{Z} \simeq (SO(2n, 1)/SO(2n))/\mathbb{Z}$ introduce additional ambiguities, e.g., how to incorporate angular potentials, and dealing with the fact that the principal series of $SO(2n, 1)$ carries an additional discrete series of representations not present in thermal $\text{AdS}_{2n+1}$. Much of the analysis for $\mathbb{H}^{2n+1}/\mathbb{Z}$ goes through for STT tensors, however, some important subtleties remain which require further study.

Another extension would be to compare to other methods of computing partition functions. For example, in [23] partition functions of free massless quantum fields on $\text{AdS}_N$ were computed using Hamiltonian techniques. The spectral data of the partition function was then encoded into a "Hamiltonian" zeta function, with the Hamiltonian being an operator acting on single particle states, and the zeta function related to the trace over the Hilbert space of the single paowrticle states. The Hamiltonian zeta function is related to single particle partition function via a Mellin transform. Given that the 1-loop determinant is related to the heat kernel via a Mellin transformation, it is expected that there is some overlap between the methods discussed here. A couple points of difference is that here we considered massive higher spin fields (mostly bosonic in character) in odd-dimensional thermal AdS, while [23] focus on massless fields that are either bosonic or fermionic in even- and odd-dimensional AdS. It would be interesting to connect these two works, especially to see how the Hamiltonian zeta function relates to the Selberg zeta function. A first step would be to see how the heat kernel computed using the method of images relates to the Hamiltonian zeta function.

A third extension to this work would be to consider non-hyperbolic spacetimes that possess sufficient symmetry, e.g., the sphere $S^N$ and the quotients $S^N/\Gamma$. In fact, it is straightforward to extend our analysis from $\text{AdS}_N$ to $S^N$ via a formal Wick rotation, where the normal modes of thermal AdS become the quasinormal modes of Euclidean de Sitter space. Understanding the relationship between the heat kernel and quasinormal mode methods of computing 1-loop determinants on $S^N$ and $S^N/\Gamma$ may lead to deeper insights into de Sitter quantum gravity. This was recently accomplished in [24] for higher spin fields in odd-dimensional Euclidean de Sitter space.

## Acknowledgments

We would like to thank Cynthia Keeler for helpful discussions. The work of VM is supported by the U.S. Department of Energy under grant number DE-SC0018330.

## A  Geometry of AdS$_{2n+1}$

One way to obtain the metric of global Euclidean AdS$_{2n+1}$ is to perform a double Wick rotation of the sphere $S^{2n+1}$ metric. Specifically, define the coordinates of the $S^{2n+1}$ sphere in terms complex numbers $(z_1, z_2, ..., z_{n+1})$ such that

$$|z_1|^2 + |z_2|^2 + ... + |z_{n+1}|^2 = 1 \,, \tag{65}$$

each with a phase $\phi_i$. For example, for $S^3$ there are two complex numbers $z_i$ with phases $\phi_1, \phi_2$; for $S^5$ there are three complex numbers $z_i$ with $\phi_1, \phi_2, \phi_3$, and so forth. It is often useful to decompose the complex numbers into real coordinates $\{x_i\}$ that embed the sphere $S^{2n+1}$ into $\mathbb{R}^{2n+2}$. For $S^5$ the three complex numbers are decomposed into six real

$$
\begin{aligned}
x_1 &= \cos\theta\cos\phi_1\,, \quad x_2 = \cos\theta\sin\phi_1\,, \quad x_3 = \sin\theta\cos\psi\cos\phi_2\,, \\
x_4 &= \sin\theta\cos\psi\sin\phi_2\,, \quad x_5 = \sin\theta\sin\psi\cos\phi_3\,, \quad x_6 = \sin\theta\sin\psi\sin\phi_3\,.
\end{aligned}
\tag{66}
$$

The corresponding line element for $S^5$ is

$$d\theta^2 + \cos^2\theta\, d\phi_1^2 + \sin^2\theta\, d\Omega_3^2\,, \tag{67}$$

where $d\Omega_3^2$ is the 3-sphere line element written in Hopf coordinates, $d\Omega_3^2 = d\psi^2 + \sin^2\psi\, d\phi_3^2 + \cos^2\psi\, d\phi_2^2$. Generalizing this procedure to $2n+1$ dimensions, the metric for $S^{2n+1}$ is

$$ds^2 = d\theta^2 + \cos^2\theta\, d\phi_1^2 + \sin^2\theta\, d\Omega_{2n-1}^2\,. \tag{68}$$

Euclidean AdS$_{2n+1}$ is now obtained by Wick rotating $\theta \to -i\rho$ and $\phi_1 \to it_E$, where $\rho, t_E \in \mathbb{R}$, and $ds \to ids$, leading to

$$ds^2_{AdS_{2n+1}} = d\rho^2 + \cosh^2\rho\, dt_E^2 + \sinh^2\rho\, d\Omega_{2n-1}^2\,. \tag{69}$$

In what follows we will relabel the phases $\phi_i$ appearing in $d\Omega_{2n-1}^2$ so that they run from $i = 1, ..., n-1$. For example, replace $d\Omega_3^2 \to d\psi^2 + \sin^2\psi\, d\phi_1^2 + \cos^2\psi\, d\phi_2^2$ in the line element for AdS$_5$.

## B  Group Theoretic Construction of the Heat Kernel

The heat kernel method [15, 16] is used to compute 1-loop functional determinants by constructing the heat kernel $K^{(s)}(x, y; t)$ with respect to the normalized eigenfunctions $\psi^{(s)}_{n,a}(x)$ of the kinetic operator $\nabla^2_{(s)}$ for a spin-$s$ field on a $d+1$-dimensional manifold $\mathcal{M}$

$$K^{(s)}_{ab}(x; y; t) \equiv \langle y, b|e^{-t\nabla^2_{(s)}}|x, a\rangle = \sum_n \psi^{(s)}_{n,a}(x)\psi^{(s)*}_{n,b}(y)e^{-tE^{(s)}_n}\,. \tag{70}$$

The subscripts $a$ and $b$ to denote the local Lorentz indices of the spin-$s$ field [16]. The 1-loop partition function $Z^{(1)}_s$ is given in terms of the trace of the coincident heat kernel $K^{(s)}(t)$

$$\log Z^{(1)}_s = \log\det(-\nabla^2_{(s)}) = -\int_0^\infty \frac{dt}{t}K^{(s)}(t)\,, \tag{71}$$

where

$$K^{(s)}(t) = \text{tr}(e^{-t\nabla^2_{(s)}}) = \int_{\mathcal{M}} \sqrt{g} d^{d+1}x \sum_a K^{(s)}_{aa}(x,x;t) \,. \tag{72}$$

When $\mathcal{M}$ is a highly symmetric homogeneous space $G/H$, group theoretic techniques can be used to write down the eigenfunctions $\psi^{(s)}_{n,a}$ of the spin-$s$ Laplacian $\nabla^2_{(s)}$ in terms of matrix elements of representations of the symmetry group. These techniques were developed by Camporesi and Higuchi [25–29] and adapted for thermal AdS by [15,16]. Here we summarize the findings of [15,16].

Euclidean AdS$_N$ is the homogeneous space $\mathbb{H}^N \simeq SO(N,1)/SO(N)$. Building the heat kernel on $\mathbb{H}^N$ requires that we construct the eigenfunctions associated with a section[10] in $SO(N,1)$. Specifically, the eigenfunctions are determined by the matrix elements of unitary representations of $SO(N,1)$ containing unitary representations of $SO(N)$. When $N = 2n+1$ these are the principal series representations[11] of $SO(2n+1,1)$ labeled by the array $R = (i\lambda, \vec{m})$ for $\vec{m} = (m_2, ..., m_{n+1})$ with each $m_i$ being a non-negative half integer, $\lambda \in \mathbb{R}$. For STT tensor fields, the array simplifies such that $\vec{m} = (s, 0, ..., 0)$. The eigenvalues $E^{(s)}$ for STT tensors is given by the quadratic Casimir

$$E^{(s)}_{R,\text{AdS}_{2n+1}} = -(\lambda^2 + s + n^2) \,. \tag{73}$$

The coincident heat kernel on $\mathbb{H}^N$ for $N = 2n+1$ can then be derived [16,28]

$$K^{(s)}(x,x;t) = \int d\mu^{(s)}(\lambda) d_s e^{tE^{(s)}_R} \,, \tag{74}$$

where $d\mu^{(s)}(\lambda)$ is the odd-dimensional generalization of the Plancherel measure, and $d_s$ is the dimension of the irreducible representation of $N-2$-dimensional sphere. When $N = 2n+1$ we have [29]

$$d^{(2n-1)}_s = \frac{s+n-1}{n-1} \frac{(s+2n-3)!}{s!(2n-3)!} \,. \tag{75}$$

Notice that for any $n$ the scalar $s = 0$ yields $d_0 = 1$, for $n = 1$ we have $d_{s>0} = 2$, and when $n = 2$, $d_s = (s+1)^2$.

The trace of the coincident heat kernel for massless spin-$s$ fields on thermal AdS$_{2n+1}$ with angular potentials $\theta_i$ is built using the method of images [16]

$$K^{(s)}(\gamma;t) = \frac{\beta}{2\pi} \sum_{k=1}^{\infty} \int_0^{\infty} d\lambda \chi_{\lambda,s}(\gamma^k) e^{tE^{(s)}_R} \,, \tag{76}$$

where the $\chi_{\lambda,s}$ is the Harish-Chandra character for $SO(2n+1,1)$

$$\chi_{\lambda,s}(\beta, \theta_1, ..., \theta_n) = \frac{e^{-i\beta\lambda} \chi^{SO(2n)}_s(\theta_1, ..., \theta_n) + e^{i\beta\lambda} \chi^{SO(2n)}_s(\theta_1, ..., \theta_n)}{e^{-n\beta} \prod_{i=1}^n |e^\beta - e^{i\theta_i}|^2} \,, \tag{77}$$

---

[10]Recall a section $\sigma(x)$ in the principal bundle $G$ of a homogeneous space $G/H$ is a map $\sigma : G/H \mapsto G$ such that $\pi \circ \sigma = e$, where $\pi$ is the projection map from $G$ to $G/H$, $\pi(g) = gH$ for all $g \in G$, $e$ is the identity element in $G$, and $x$ are coordinates on $G/H$.

[11]For even-dimensional AdS$_{2n} \simeq (SO(2n,1)/SO(2n))$ the principal series of $SO(2n,1)$ carries an additional discrete series of representations not present in the odd-dimensional case [16]. It turns out, however, that for STT tensor fields, this additional discrete series does not contribute for $n > 1$ [28]. The odd-dimensional analysis is then readily extended to even-dimensional thermal AdS$_{2n}$ for $n > 1$, when all of the angular potentials are switched off. The only change to the functional determinant in (78) is that, the dimension $d_s$ (75) must be replaced with its even-dimensional counterpart.

with $\chi_s^{SO(2)}$ is the character of $SO(2n)$ in the $s$ representation. The group element $\gamma$ in (76) $\gamma = e^{i\beta Q_{12}} e^{i\theta_1 Q_{23}} ... e^{i\theta_n Q_{2n+1,2n+2}}$ with $Q$'s as generators of $SO(6)$. When all of the angular potentials are turned off, $\theta_i = 0$ for all $i$, the character $\chi_s^{SO(2n)} = d_s$ in (75).

The functional determinant of the Laplacian of a massive spin-$s$ field on $\mathbb{H}^{2n+1}/\mathbb{Z}$ is then given by

$$-\log\det(-\nabla^2_{(s)} + m_s^2) = (2 - \delta_{s,0}) \int_0^\infty \frac{dt}{t} K^{(s)}(\gamma; t) e^{-tm_s^2}$$
$$= (2 - \delta_{s,0}) \sum_{k \in \mathbb{Z}_+} \frac{2}{k} \frac{e^{-\beta k(\Delta_s - n)}}{e^{-nk\beta} \prod_{i=1}^n |e^{k\beta} - e^{ik\theta_i}|^2} \chi_s^{SO(2n)}(k\theta_1, ..., k\theta_n) \,, \tag{78}$$

where we have introduced the conformal dimension

$$\Delta_s = n + \sqrt{s + n^2 + m_s^2} \,. \tag{79}$$

When $n = 1$ (78) reduces to the functional determinant for arbitrary spin-$s$ fields on thermal AdS$_3$ [15], where $\chi_s^{SO(2)} = \cos(2\pi s k \tau_1)$, with $2\pi\tau_1 = \theta_1$, and $2\pi\tau_2 = \beta$.

It is beneficial for us to put the functional determinant (78) in a way exemplifying its modular form. Specifically, using (46) and (47), it is straightforward to rewrite (78) as

$$-\log\det(-\nabla^2_{(s)} + m_s^2) = (2 - \delta_{s,0}) \sum_{k=1}^\infty \frac{2}{k} \frac{\prod_{j=1}^n (q_j \bar{q}_j)^{k\Delta_s/2j}}{\prod_{i=1}^n |1 - q_i^k|^2} \chi_s^{SO(2n)}(k\theta_1, k\theta_2, ..., k\theta_n) \,. \tag{80}$$

In particular, for AdS$_5$ we have

$$-\log\det(-\nabla^2_{(s)} + m_s^2) = (2 - \delta_{s,0}) \sum_{k=1}^\infty \frac{2}{k} \frac{(q_1 \bar{q}_1)^{k\Delta_s/4} (q_2 \bar{q}_2)^{k\Delta_s/4}}{|1 - q_1^k|^2 |1 - q_2^k|^2} \chi_s^{SO(4)}(k\theta_1, k\theta_2) \,, \tag{81}$$

where, since $SO(4) \simeq SU(2) \times SU(2)$, the character $\chi_s^{SO(4)}$ is given by

$$\chi_s^{SO(4)}(k\theta_1, k\theta_2) = \chi_s^{SU(2)}(\alpha_+) \chi_s^{SU(2)}(\alpha_-) = \frac{\sin[(s+1)\alpha_+]}{\sin(\alpha_+)} \frac{\sin[(s+1)\alpha_-]}{\sin(\alpha_-)} \,, \tag{82}$$

with $\alpha_\pm \equiv (\theta_1 \pm \theta_2)/2$.

Let us briefly verify that our expression for the 1-loop partition function in terms of Selberg zeta functions for general spin in thermal AdS$_5$ (59) is equal to the expression found using the heat kernel and the method of images (78). We do this by first writing the Selberg zeta function $Z_\mathbb{Z}(z)$ in (43) for $n = 2$:

$$Z_\mathbb{Z}(z) = \prod_{k_1, ... k_4 = 0}^\infty \left[1 - e^{i\theta_1 k_1} e^{-i\theta_1 k_2} e^{i\theta_2 k_3} e^{-i\theta_2 k_4} e^{-\beta(k_1 + k_2 + k_3 + k_4 + z)}\right] \,. \tag{83}$$

Note then that

$$\sum_{m,m'=0}^s \log Z_\Gamma(z) = \sum_{m,m'=0}^s \sum_{k_1, ..., k_4 = 0}^\infty \log(1 - x) = -\sum_{k=1}^\infty \sum_{m,m'=0}^s \sum_{k_1, ..., k_4 = 0}^\infty \frac{x^k}{k} \,, \tag{84}$$

with

$$x \equiv e^{i(\theta_1 - \beta)k_1} e^{-i(\theta_1 + \beta)k_2} e^{i(\theta_2 - \beta)k_3} e^{-i(\theta_2 + \beta)k_4} e^{-\beta\left[\Delta_s + \frac{i\theta_1}{\beta}(m+m'-s) + \frac{i\theta_2}{\beta}(m-m')\right]} \,, \tag{85}$$

where we made the replacement $z = \Delta_s + \frac{i\theta_1}{\beta}(m + m' - s) + \frac{i\theta_2}{\beta}(m - m')$. Performing the geometric series over each $k_i$, $i = 1, ..., 4$, and identifying the sum over $m$, $m'$ as the character $\chi_s^{SO(4)}(k\theta_1, k\theta_2)$ (58), we are left with (78) in the case of $n = 2$ (upon implementing the $(2 - \delta_{s,0})$ we left of above, and picking up a factor of 2 from the relation between $\log Z_s^{(1)}$ and $\log\det(-\nabla_s^2 + m_s^2)$), We recognize the remaining sum over $k$ as the sum over images.

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
