# Peer review of "Normal modes in thermal AdS via the Selberg zeta function"

_SciPost Physics, doi:SciPost Phys. 9, 009 (2020)_

## Round 1 · Referee Report · Anonymous (Referee 1) · 2020-5-19

Report

The present paper builds on previous work done by the authors in with C. Keeler (refs 9 & 12 in the paper) and presents expressions for the partition function of arbitrary spin fields in quotients of odd dimensional AdS spaces, namely thermal AdS, with and without additional chemical potentials.

The paper is well written, presented and organized and the results are mostly clearly understandable.

The results are interesting and worthy of publication, once some points are addressed.

Requested changes

  1. Since there are heat kernel results already available in quotients of AdS in even dimensions, and in AdS the presence of fluxes, we would request the authors to comment on the possibility of using their methods to arrive at the partition function in those cases.

  2. It would help comparison with previous work if at least for the case AdS5, the authors were to write their expressions (3.25) and (3.26) in a manner that may be readily compared with their expression (B.9), a sum over images in AdS5.

  3. It is noted below their expression (3.27) that the arguments of the Selberg zeta function arise from a careful removal of the non-square integrable zero modes. In our opinion, the paper might benefit with some details being provided about this procedure, since the jump from AdS3 to arbitrary odd dimensional AdS spacetimes is so far made quite rapidly.

  4. We also invite the attention of the authors to the interesting paper https://arxiv.org/abs/hep-th/0606186 where computations of these partition functions were carried out using Hamiltonian techniques.

  • validity: good
  • significance: good
  • originality: good
  • clarity: good
  • formatting: perfect
  • grammar: perfect

Author:  Andrew Svesko  on 2020-05-19  [id 828]

(in reply to Report 1 on 2020-05-19)

We thank the referee for their constructive and thought-provoking feedback. Below we will address their editorial suggestions.

The first point the referee brings up is the request we discuss on the possibility of using our methods to study partition functions on quotients of even dimensional AdS. Already in the draft we briefly discuss the possibility to extend to even dimensions for the scalar field (see footnote 7) and also how the even-dimensional case may hold for STT tensors via the heat kernel method (footnote 11). In the case of higher spin fields, even dimensional quotients H^{2n+1}/Z present additional ambiguities, in particular how to incorporate the angular potentials, and how to deal with the fact that the principal series of SO(2n,1) carries an additional discrete series of representations not present in the odd-dimensional case. Similar difficulties arise in the heat kernel method, as discussed more throughly in (1103.3627). We had added a paragraph to the Discussion section to highlight these ambiguities and leave this extension for future work.

The second point is a suggestion to show more directly how our 1-loop determinant via Selberg zeta functions in Eq. (3.25) relates to the heat kernel/image sum method in Eq. (B.9), at least in the case of AdS_{5}. We agree this would be useful and illustrative of our result. As such we have added a paragraph in Appendix B just below Eq. (B.13), where provide the more intricate details relating Eq. (3.25) to Eq. (B.9) in the case of AdS_{5} for an arbitrary higher spin field.

Thirdly, the referee requests additional details be given concerning the removal of non-square integrable zero modes. Prior to our edits, we provided some of this background in the context of the BTZ black hole in the paragraph below Eq. (2.24). We have expanded on this discussion for the BTZ case, and how the analysis almost trivially extends to thermal AdS_{3}. Moreover, when extending to the higher dimensional quotients we clarify that we \emph{expect} an analogous computation be completed, and remark that accomplishing this would require a full analysis of the quasinormal mode method of computing partition functions of higher spin fields on higher dimensional AdS backgrounds, an analysis that has not yet been completed in the literature -- indeed knowledge of the higher spin normal mode frequencies are required; these frequencies were predicted in our article. We mention how such an analysis is lacking, though it would be worthwhile to study in completeness. We include these clarifications in the paragraph just below Eq. (3.27).

Finally, the referee suggests we examine the article (0606186), where partition functions on AdS are computing using Hamiltonian methods, and comment on how it might relate to our techniques. We include a paragraph in the conclusion briefly mentioning the connections (and differences) between the Hamiltonian method, the heat kernel method and our connection with the Selberg zeta function; specifically how the Hamiltonian zeta function relates to the Selberg zeta function.

We hope that these clarifications will adequately satisfy the referee's request.

---

## Round 2 · Referee Report · Anonymous (Referee 1) · 2020-7-9

Report

We thank the authors for considering our comments, and are happy to recommend the manuscript for publication in SciPost.

---

## Round 2 · Author Response

See referee reply posted online.

---

## Round 2 · List of Changes

-- Added paragraph to discussion to further clarify the ambiguities with the even dimensional analysis.
-- We have added a paragraph in Appendix B just below Eq. (B.13), where provide the more intricate details relating Eq. (3.25) to Eq. (B.9) in the case of AdS_{5} for an arbitrary higher spin field.
-- We have expanded the discussion concerning the analysis of removing non-square integrable zero modes for higher spin fields in both the BTZ case and thermal AdS_{3}. We further clarify what would need to be shown for higher dimensional AdS solutions, but explain why we think the analysis is possible.
-- Added paragraph to discussion comparing our methods of computing the partition functions to alternative methods, e.g., using Hamiltonian methods.

---

## Editorial Decision

published